# Penal and Custodial Control of Female Criminality in Spain from a Gender Perspective

## María Acale Sánchez

Department of Public, Criminal and Procedural International Law, University of Cadiz, 11405 Jerez, Spain; maria.acale@uca.es

**Abstract:** Gender is a uniquely important factor in women's lives and female criminality alike: A study of statistics on female criminality worldwide shows that several gender-related factors may determine a woman's commission of an offence, her life in prison, and perhaps even her future. Reflecting on this problem based solely on official statistics is difficult because some variables remain invisible from this perspective. Consequently, it is necessary to study the problem by applying criminological methods and examining past judgements. Whether it be an analysis of offences predominantly associated with women in the statistics (drug trafficking), or of other crimes that are not (terrorism, white/pink-collar crime), it is possible to identify gender patterns that in micro-criminological terms serve to elucidate the reasons why these women have committed an offence. It may even be possible to prevent female criminality by tackling the gender stereotypes that are present in all these crimes.

**Keywords:** gender perspective; female criminality; prison treatment

## 1. Introduction

The imprisonment of female offenders is associated with a number of specific issues that occur throughout the world, and Spain, Italy, or the United States are no exception. This has been recognised by the United Nations, which in response to data indicating an increase in the female prison population, published the Resolution of 16 May 2011 containing the "United Nations rules for the treatment of women prisoners and non-custodial measures for women offenders", more commonly known as the Bangkok Rules.

The study of the incarceration of women is considerably enriched by a dynamic analysis of such women's entire lives. In this broader context, it can be seen that there are a number of gender-related factors in female criminality that may have determined commission of the offence, that exert a strong influence on a woman's life in prison and that will also condition her return to society: These factors have held her "prisoner" without her realising it long before she became subject to the formal control of criminal law. Consequently, an analysis of the crimes committed against women and yet that very few women themselves commit may yield an instrument that serves to prevent female criminality and victimisation alike.

At the international level, this problem is not due to the fact that the number of crimes committed by women is very high; on the contrary, the statistics indicate less female than male criminality. To date, this circumstance has aroused little attention; however, precisely for this reason, a study of the phenomenon of female criminality is of great value because qualitatively, it is brimming with information on why women commit crimes. Thus, whether it be an analysis of offences predominantly associated with women in the statistics (drug trafficking), or of other crimes that are not (terrorism, white-collar crime), it is possible to identify gender patterns that in micro-criminological terms explain the reasons why these women have committed an offence. Although it may not be possible to draw

conclusions from these patterns in macro-criminological terms, this should not hinder identification of the gender norms common to all of them.

This overview of female offenders' lives evidences that social reintegration is harder for women than for men because they must overcome more obstacles while imprisoned. In this respect, following the adoption of the Bangkok Rules on 21 August 2013, the General Assembly published a report submitted by the General Rapporteur entitled "Pathways to, conditions and consequences of incarceration for women", which stated that "stigmatisation, social alienation and feelings of shame and guilt may also contribute to hindering reintegration. Stigma and the loss of certain rights are a challenge to re-establishing relationships and social ties. Family and community support is critical to successful reintegration, and also to decreasing the chance of recidivism" (UN 2014, p. 22).

Below, I present the results obtained from a study in Spain on female criminality, the first phase of which was carried out in 2010 and 2015 (Acale Sánchez 2017) and encompassed a statistical and legal analysis, surveys, and focus groups. The second phase of the study, reported here, concerns an analysis of three forms of criminality. According to the statistics, the first of these, drug trafficking, is the main crime for which women are imprisoned worldwide. In contrast, the number of women convicted of terrorism and white-collar crime is negligible, and consequently, such crimes barely appear in the statistics. Nevertheless, considerable attention is paid to them here because they provide illuminating qualitative data. My goal is to demonstrate that despite the low numbers involved, the women who commit these crimes are all conditioned by the same factor: The patterns of gender relations that persist in modern societies.

The data presented here is probably applicable to other societies and countries, because regardless of their individual penal codes or custodial legislation, all societies are governed by a common law: Patriarchy.

The reason for selecting these types of crime is that nowadays, terrorism, drug trafficking, and white-collar crime are all committed on an international stage. Thus, such crimes are often initiated in one country, but conclude in another, generating not only a crime, but also a punishment without borders, transcending domestic law.

Added to this is the growing phenomenon of international migration. In consequence, a snapshot of the female population in Spanish prisons shows a multicultural and multinational population. While this may be a static image of Spanish prisons, it nevertheless reflects the situation of women of different nationalities, because a multitude of countries are represented inside Spain's prisons.

This special issue of the journal, *Social Sciences*, provides the data necessary for readers to carry out a comparison, and they are invited to do so. Such is the aim of this paper: To report on the current situation of female criminality in Spain, which in its essentials can probably be extrapolated to any other country that shares the same culture and customs.

## 2. The Impact of Gender on Crime

Crime statistics and data show that, as has been the case for centuries, women commit fewer and less serious crimes than men (Islam et al. 2014; Medlicott 2007). However, there has been a much greater increase in female than male criminality over the past 20 years (Almeda Samarach 2007). The factors that explain the reasons why women are committing crimes more often than before (Davies 1999)—albeit still much less frequently than men—seem to be related to women's increasingly active role in society and in the family; from being protected members, women have become super-protectors, and now often occupy the role of "head" of the family traditionally reserved for men. In view of the economic problems that this entails, due in part to women's lack of professional training, which compels them to work in sectors that receive the least social recognition, are the worst paid, and which bear the brunt in times of crisis, female employment is necessarily linked to the feminisation of poverty and in consequence, to the emergence of associated forms of criminality. It is precisely this criminality that receives harsh treatment under criminal law and often leads to a custodial sentence (Bodelón González 2007). Furthermore, female criminality has aroused scant research attention, and what little

it has received is often interpreted in male terms (Larrauri Pijoán 1994; Acale Sánchez 2006; Islam et al. 2014; Heidensohn and Silvestri 2012) (Tables 1 and 2).

**Table 1.** Types of offences for which women receive custodial sentences.

| Crime | Number |
|---|---|
| Drug trafficking | 1552 |
| Against property | 1383 |
| Homicide | 298 |
| Assault | 188 |
| Remainder | 575 |
| Total | 3996 |

Source: by the author based on Spanish Ministry of the Interior data.

**Table 2.** Types of offences for which men receive custodial sentences.

| Crime | Number |
|---|---|
| Against property | 18,803 |
| Public health | 10,361 |
| Gender-based violence | 3841 |
| Homicide | 3594 |
| Sexual assault | 3050 |
| Remainder | 8790 |
| Total | 48,449 |

Source: by the author based on Spanish Ministry of the Interior data.

A comparison of the offences for which men and women are imprisoned reveals different preferences according to gender. Thus, an analysis of Spanish Ministry of the Interior data on the Spanish prison population on 31 December 2015 and the specific offences that gave rise to custodial sentences for women indicates that crimes against public health (drug trafficking) occupied first place, followed by crimes against property, homicide, and assault. The remainder appeared in less significant numbers (for the data on 31 December 2010, see Acale Sánchez 2011).

Hence, in view of the two categories that accounted for the highest number of custodial sentences, female criminality can be characterised as being directed towards the ultimate goal of obtaining money, either to meet the needs arising from women's dependence on the very drugs they traffic on a small scale, or to alleviate the poverty in which they and their dependent family members live. Female criminality is therefore a function of their deprivation or financial distress; hence also the specific health problems they experience (e.g., OSDI) (Igareda 2007). The category of "remainder" includes a multitude of offences that although numerically insufficient to merit being listed individually alongside the other unquestionably more serious crimes, present a very high qualitative value, such as convictions for terrorism or white-collar crimes (or rather, in this case, for pink-collar crimes), and are harshly stigmatised due both to gender and the ramifications of incarceration.

The distribution by category changes in the case of men.

In the case of men, it can be seen that crimes against property now occupy first place in the criminal activity ranking, and although no specific mention is made in this case of the kind of offences involved, there is no doubt that these include both violent and non-violent crimes against property. Meanwhile, drug trafficking now occupies second place. Hence, a comparison with crimes that led to custodial sentences for women reveals a shift in predominant criminal activities. It is striking that the above two categories of crime are followed by another one meriting a category of its own, gender-based violence, convictions for which currently account for 7.6% of the male prison population. Sexual assault, which appears after homicide, also merits its own category. This accounts for the negligible number of 54 women in prison, and the statistics do not reveal if these were the sole perpetrators or acted in complicity with a man or if they sexually assaulted a minor, an intellectually disabled person, or a person in full possession of their mental faculties (see Acale Sánchez 2017).

Irrespective of the above, the gradual decline in male criminality contrasts strikingly with the maintenance of female criminality. To some extent, it could be argued that the number of crimes committed by women is not dropping at the same rate as the number committed by men because the reasons that determine male and female criminality are different. The rationale for this interpretation is that female criminality is the result of structural causes that persist over time, whereas male criminality is not explained by structural reasons inherent to the male condition, but rather by other factors that are more easily identifiable and, therefore, controllable.

An analysis of inmate age reveals that the most common age band for men and women alike is 41–60 years old. This implies that the correctional facilities that house them must be equipped to accommodate an adult prison population (See date about Unite State in Mallicoat and Ireland, 2014). According to data for the 31 December 2015, the age distribution in Spanish prisons was as follows in Table 3.

The figures for young men and women and for men and women aged over 60 years are proportionally similar. This information requires analysis, because the wide disparity between male and female criminality occurs in the central stage of life, whereas in adolescents and young and older adults, both sexes appear to tend proportionally to present similar, albeit not identical, trends. Gender roles unquestionably exert more influence during these central years of life, when women bear children and people have family responsibilities.

The age at which most differences exist in general terms is the stage when women have greater family commitments and assume responsibility for caring for the dependent members of their family. Hence, even today, gender roles still continue to govern women's lives. In this respect, it should be underscored that women imprisoned at this stage of their lives experience guilt at failing to fulfil their responsibilities that is not felt in equal measure by men, who because of their gender have not assumed care of the family beyond providing economic sustenance, from which women are not exempt either. Hence, a custodial sentence entails additional suffering for women (Acale Sánchez 2017).

**Table 3.** Prison inmate population by age group and sex.

| Age | 2009 | | 2010 | | 2011 | | 2012 | | 2013 | | 2014 | | 2015 | |
|---|---|---|---|---|---|---|---|---|---|---|---|---|---|---|
| | M | F | M | F | M | F | M | F | M | F | M | F | M | F |
| 18–20 | 615 | 40 | 651 | 24 | 581 | 25 | 510 | 29 | 449 | 32 | 347 | 14 | 305 | 12 |
| 21–25 | 6.654 | 550 | 6.623 | 467 | 6.625 | 424 | 5.372 | 368 | 5.163 | 346 | 4.703 | 343 | 4.187 | 281 |
| 26–30 | 11.322 | 984 | 11.206 | 933 | 10.339 | 843 | 8.305 | 676 | 8.120 | 637 | 7.922 | 592 | 7.383 | 571 |
| 31–40 | 19.189 | 1.576 | 19.214 | 1.520 | 18.414 | 1.372 | 18.108 | 1.386 | 17.933 | 1.467 | 17.383 | 1.446 | 16.455 | 1.401 |
| 41–60 | 15.737 | 1.379 | 15.711 | 1.409 | 15.761 | 1.300 | 18.009 | 1.547 | 18.569 | 1.560 | 18.758 | 1.609 | 18.518 | 1.605 |
| +60 | 1.378 | 94 | 1.388 | 105 | 1.651 | 105 | 1.622 | 113 | 1.704 | 118 | 1.880 | 111 | 1.852 | 136 |
| N record | 52 | | 9 | | 5 | | 0 | | 5 | | 1 | | 5 | 0 |

Source: by the author based on Spanish Ministry of the Interior data.

## 3. Qualitative Analysis of Female Criminality

### 3.1. Women Drug Traffickers

Prison statistics show that the largest group of female prisoners comprises women convicted for drug trafficking offences (see Mallicoat and Ireland 2014). This is corroborated by an analysis of court rulings, which indicates that criminal organisations use women as "mules" who smuggle small amounts of drugs, in their own bodies, into Spain (for a description of the process of ingestion, trip, and arrival in Spain (see Dorado María 2005, 317f.; see also Bodelón González 2007). Small scale drug trafficking is an offence that neither requires any special skill nor poses any personal risk to the offender, and whose detection or not by the police is often a matter of random chance. It seems that these reasons persuade women to commit this offence (whether or not they consume these substances)[1] (Puente Aba 2012, 97ff.; González Agudelo 2015).

The statistics suggest—although they do not confirm—that these crimes are usually committed individually rather than in concert with others, even though the women are often no more than the last link in the chain of an organisation run by criminals or by a husband or lover who has set them on the path to crime (Naredo Molero 2005). Furthermore, the women are not members of the organisation; instead, they simply work for it, and if they are caught, they tend not to inform on other members in the hope that the organisation will honour its "debt" to them at some point in the future, for example, by "hiring" them anew once they have been released and are again free to participate in its activities. Consequently, although the Penal Code provides for a reduced sentence if the perpetrator collaborates with the judicial system (art. 376 of the Penal Code), many victims do not benefit from this because they prefer to "protect" the organisation, whether out of fear or due to "solidarity".

An analysis of the nationality of these women might erroneously link female drug trafficking to illegal immigration, since many of the women imprisoned for this crime are foreign. However, a more careful examination reveals that most foreign female convicts are held in prisons in Andalusia and Madrid because the vast majority of them had attempted to smuggle hashish and cocaine into Spain via the Strait of Gibraltar or the Adolfo Suárez Madrid-Barajas airport, indicating that many of these foreign women were not resident or domiciled in Spain before committing their crime and were not therefore "immigrants" (Aguilera Reija 2005; Boza Martínez 2015). Thus, a study of rulings indicates that foreign women with tourist visas are detained at the airport for transporting drugs inside their bodies or in their luggage, before they have even left the international arrivals area. Alternatively, having detected the operation, the police may allow them to leave the airport while monitoring their movements to identify the other people involved. Although these women are foreign, they are not immigrants in Spain; therefore, when computing crimes committed by foreign women, it should be borne in mind that strictly speaking, these cases should not be considered crimes committed by immigrants because they are not (Boza Martínez 2015). As a result, they do not have ties or a domicile in Spain and will therefore spend more time in prison, because all the mechanisms provided for in Spanish prison legislation aimed at social reintegration are based on the assumption that people will be reintegrated into the society in which they lived before, and this is not the case.

---

[1]　This information can also be analysed in judgements, which include some rulings that from a gender perspective, attempt with greater or lesser success to take into consideration the mitigating circumstances surrounding perpetration of an offence. One example that merits such an analysis is the ruling of the Provincial Court of a Coruña 7/2012, of 10 February (JUR/2012/97742), convicting a man and a woman—a couple—for the crime of drug trafficking. For an identical crime, each was given a different sentence, which was more lenient in the case of the woman based on the following reasoning: "The decision to impose upon the accused a custodial sentence close to the legal minimum is based on the fact that she is the mother of two young daughters, she is unemployed and her husband consumes drugs". The husband's sentence was also reduced, but in this case due to the mitigating circumstance of drug addiction. Despite the court's commendable attempt to impose the minimum possible sentence on both, it should be noted that maternity, employment, or living with a drug addicted husband do not add anything to the drug trafficking offence committed and are completely unconnected with the same; therefore, strictly speaking, this interpretation of the law should be avoided.

Due to the roles attributed to women by patriarchal society, many of them try to protect other members of their family (husbands, sons, daughters, fathers, mothers) by remaining silent and not informing on them. For example, following a police search, they may take responsibility for a crime committed by another, who in most cases will have a criminal record and would receive a more severe sentence due to recidivism, whereas these women "foreign" to the world of crime will by definition not have a criminal record and would thus receive a more lenient sentence.

In this respect, it should be borne in mind that although article 23 of the Penal Code considers the relationship between the victim and the perpetrator of the crime a reason for exacerbating or mitigating the sentence, the perpetrator's relationship with other persons implicated in the crime is irrelevant from the standpoint of criminal law, although it has an undeniable importance from the standpoint of criminology. In consequence, courts analyse the imputation of criminal liability very carefully in these cases, since people living at the same domicile cannot all automatically be considered perpetrators, for example. In this respect, the ruling of the Provincial Court of Cadiz 432/2009, of 27 November (ARP/2012/1492), seems to assume a kind of family or "family clan" responsibility, although the mother was eventually acquitted. Fleeing from simplistic interpretations, many judgements attempt to restrict the imputation of criminal liability. Thus, the ruling of the Provincial Court of Alicante 75/2012, of 21 March (JUR/2012/294764), analyses the criminal responsibility of the wife of a man convicted of drug trafficking, and states that: "The presence, in her husband's possession, of a considerable quantity of drugs and of the implements previously referred to, in her domicile, is evidence of crime. Nevertheless, having convicted Faustino, there remains the more favourable hypothesis for the accused that he alone was responsible for drug trafficking".

By the same token, there have been several cases where a woman has been convicted of drug trafficking after attempting to smuggle drugs into prison for her son or husband, such as the ruling of the Provincial Court of Barcelona 12/2012, of 9 January (JUR/2012/88078), convicting a Moroccan woman. In these cases, the processes of victimisation of the woman and criminalisation of her behaviour occur simultaneously.

Lastly, several sentences have taken drug abuse and drug addiction on the part of the female drug trafficker into consideration, most of the time considering the circumstance of drug addiction provided for in article 21.7, and much less frequently applying the incomplete defence of drug addiction established in articles 21.1 and 20.2. What is striking from this perspective is that these were Spanish women who in all cases were trafficking drugs as a means to fund their own dependence. This indicates that in the case of Spanish women, their offences are usually functionally related to their drug addiction and prevention should focus on treatment and rehabilitation, whereas in the case of foreign women, their offences are functionally unrelated to any kind of drug dependence.

They will spend a long time in prison because drug trafficking offences are punishable by harsh custodial sentences. It should be noted, however, that the reform of the law on drug trafficking introduced by Organic Law 5/2010 partially alleviates these figures, since it includes a discretionary mitigation of the sentence in the new second paragraph of article 368, which establishes that "notwithstanding the preceding paragraph, the courts may impose a lower sentence than that indicated in light of the limited entity of the fact and of the personal circumstances of the perpetrator". As we shall see, Spanish prison legislation still contains provisions that tend to shorten prison sentences "by reason of sex", provisions that have no other grounds than the very gender stereotypes that may have led to these women to commit a crime.

### 3.2. Women Convicted of Crimes of Terrorism

Although terrorism is not specifically identified in the statistics, from a legal and qualitative point of view, it is interesting to analyse the characteristics of women convicted of terrorism (Pérez Sedeño 2012), one of the most serious crimes punishable under the Penal Code following the adoption of Organic Law 1/2015, with a reviewable life sentence.

For many years, the women convicted of these crimes in Spain were primarily female members of the terrorist group, ETA (and also GRAPO)[2], an organisation that according to the facts reported in judgements, treated all its members alike, entrusting women with operations of the same gravity as those perpetrated by men (assassination or kidnapping). Equally, they received the same prison treatment: Dispersion, classification as first-degree [high-risk] prisoners, and a special detention regime, policies that have been specifically designed to separate terrorists from each other. Now that ETA has disbanded, there is an urgent need to review this custodial policy, because it no longer serves any purpose now that the organisation it was intended to destroy by isolating its members has ceased to exist[3].

As a result of these conditions, which constitute a substantial impediment to maintaining family and friendship ties, women who spend years of their lives locked up and distant from their loved ones suffer much more from the stigma attached to prison because they are simultaneously subject to gender stereotypes. At the same time, it should be borne in mind that gender is also a factor in their social reintegration.

This was reflected in the press a few months ago, when reporting the situation of a woman convicted of crimes of terrorism who was sent to prison in 2005 to serve a 13-year sentence. During her sentence, she gave birth to a girl who stayed with her in prison until she was three years old. Just before the girl's birthday, on one of the weekends when she was staying with her father, he tried to kill her, inflicting very serious injuries on her. Following this event, the prisoner presented a written apology, repenting her crimes, asking for the victims' forgiveness, saying that all that mattered to her was her daughter, and asking that she be allowed to be with her daughter given the serious circumstances. The prison authorities granted her request and she was awarded second-degree [ordinary] status pursuant to art. 100.2 of the Prison Regulations, under which execution of the sentence can be adapted to the characteristics of a particular case. The case attracted considerable media attention (shocked by the violence inflicted on to the girl, but bearing in mind that her mother was a convicted terrorist), but nonetheless the main fact went unnoticed: The mother had entered prison in 2005 to serve a 13-year sentence. Consequently, at the time of requesting reclassification, she had already served three-quarters of her sentence and therefore should have been awarded third- [semi-custodial] rather than second-degree status if she had complied with all the other special requirements established by penal legislation for prisoners convicted of terrorism offences, which are essentially based on a normative concept of good behaviour. Neither did public opinion question why this woman had spent all those years incarcerated as a first-degree prisoner subject to a high security regime and therefore furthest from the exercise of freedom (Acale Sánchez 2018).

These days, the women convicted of terrorism and held in maximum security prisons in Spain belong to jihadist terrorist organisations[4], and the first question to ask concerns the role this organisation grants women. A supposed Islamic State that kidnaps girls for months and uses them

---

[2]  Some of the most famous female members of this terrorist organisation were Inés del Río (who appealed the Parot doctrine to the European Court of Human Rights, which in its judgement of 21 October 2013 ruled that its application was contrary to the Convention on Human Rights because it violated the principle of legality) and Dolores Gonzalez Katarain (also known as Yoyes), a woman convicted of terrorism who wished to leave ETA and was subsequently assassinated by the organisation itself.

[3]  For all of them, see Cuerda Riezu (2008).

[4]  Thus, for example, the importance in the Penal Code of the principle of minimum intervention is somewhat diminished in the punishment of glorification of the crime and humiliation of the victims as a terrorism offence in articles 578 and 579, which have generated a non-homogeneous body of judgements in which harsh prison sentences are imposed on people who base their defence on their fundamental right to freedom of ideology and freedom of expression guaranteed in articles 16 and 20.1 of the Spanish Constitution. These crimes are defined by indeterminate elements, which demand a judicial interpretation of what is meant by "glorification", "justification", "infamy", "contempt", "disdain", or "humiliation". The diversity of judicial interpretations of these core elements in Spain should prompt our legislators to rethink their wording, mainly because if the Constitution enshrines the right to freedom of ideology and religion, criminal law cannot punish the exercise of these fundamental rights, but should become an instrument that instead guarantees their exercise. Hence, criminal law should be a minimum criminal law, because every time that it defines an act as a crime, it restricts freedoms. Notable among the various sentences that have recently been handed down are the National Court Sentence 15/2016, of 16 November 2016; the

as human shields is unlikely to be much more respectful of women when they belong to these organisations. There is some evidence to suggest that women and/or young people are used to commit lesser attacks, ensuring that the more difficult operations are left in the hands of men.

The reform of terrorism offences established by Organic Law 2/2015 included acts whose criminal nature is clearly questionable: As a result, Spanish prisons now hold women convicted of acts, which strictly speaking, according to the principle of criminal law, should not even be considered as constituting an attempted crime, but since they have been expressly categorised, they are not covered by the protection of legal interests that the Code safeguards in general terms. These women are therefore doubly victimised because they live in a society that discriminates against them and are also used by criminal organisations to commit offences that will achieve the latter's strategic or personal goals.

When they arrive in prison convicted of these crimes, they are subject to a very harsh prison regime heavily influenced by the Penal Institution Circular I-02/2016, of 25 October 2016, which contains a framework programme for intervention with Islamist prisoners considered "violently radicalised"[5], the purpose of which is not to offer a treatment programme, but to ensure the internal security of the correctional facility. Hence, the consent of jihadist prisoners is irrelevant[6]. Pursuant to the Circular, these people are subjected to a punitive regime with no regard for their social reintegration.

The punitive nature of their prison sentence is twofold: Not only are they deprived of liberty in a harsher prison environment, but the regime itself also prevents many inmates from abiding by their customs. In particular, this is reflected in the Ruling of the National Court, Criminal Division, Section 1, 530/2017, of 17 July 2017, Rec. 195/2017, whereby a woman was refused the possibility of wearing the veil in prison, citing security reasons based on an internal prison regulation prohibiting inmates from wearing clothing that could prevent their identification (for all, see Cuerda Riezu 2008).

It is clear that rather than encouraging them to question the gender stereotypes that may have determined their exploitation by jihadist terrorism, the prison regime to which these women are subjected prompts them to identify more strongly with their group, which does not repudiate them or prevent them from abiding by their customs.

### 3.3. Women Convicted of White—or "Pink"—Collar Crimes

Prison sentences in Spain for the first women convicted of white-collar crime have been doubly conditioned by the fact that from the point of view of criminal law, such acts have not always been punished with imprisonment, nor is it easy to identify those responsible, to the extent that due to the complexity of these offences, they often remain undetected by the police. Furthermore, white-collar crime has traditionally—although not exclusively—been a very male crime, because it is committed in occupational and professional fields that for reasons of gender exclude women or relegate them to lower positions with little of the capacity for action required for this complex crime (Vassallo Sambuceti 2011).

White-collar crime was first defined by Sutherland in 1949, who used the symbol of a white collar and tie to distinguish such offences from blue-collar crime, committed by those who lack a similar status or economic means (Sutherland 1949). By focusing attention on a white collar and tie, he also endowed this crime with a sex: Men are the subject of this field of criminology, not women, whose

---

National Court Sentence 3/2018, of 2 March; the High Court Sentence 95/2018, of 26 February 2018; and the ruling of the Central Court of Instruction of 28 June 2016.

[5]  Which in turn refers to the Directives on correctional facilities and probation in the case of radicalisation and violent extremist offenders, approved by the Committee of Ministers of the European Commission on 2 March 2016, in which not a sole express reference is made to terrorism.

[6]  According to Nistral Burón (2017), prisons are aware that "the aspects inherent to the jihadist movement—religion, proselytism, cultural values, etc.—render it difficult to accurately determine the state of this phenomenon inside prison facilities, especially given that these inmates know they are subject to close surveillance and therefore modify their attitudes and behaviours to not draw the attention of prison staff". Contrary to the author's claims, such chameleon-like behaviour should not be criticised as a self-defence technique in the face of non-consensual surveillance of inmates with more far-reaching goals.

sex has prevented their entry into the business world; thus, cloistered in their homes by their criminal white-collar husbands, they have been unable to commit business-related offences (Davies 1999).

However, women's gradual incorporation into the business world has been accompanied by an increase in female convictions for economic crimes and corruption. A scientific study of this kind of crime requires a combination of approaches: In addition to an analysis of the criminological premises of white-collar crime, a gender perspective must be applied to address a new category of offence, "pink-collar crime".

An analysis of judgements reveals that pink-collar crime is more strongly influenced by gender stereotypes than other areas of female criminality, as indicated by some of the sentences handed down in recent years in Spain. The first of these is the judgement of the Supreme Court of 23 June 2014 (concerning "Operation Malaya"), which, among other people, punished two very well-known women in the world of celebrities for money laundering offences, because they had helped to launder the illicit gains obtained by one of the ringleaders of a criminal organisation, with whom both women had maintained an intimate relationship. The second is the judgement of the National Court of 23 May 2018 (the "Gürtel Case") punishing multiple people for a large number of crimes of embezzlement, perversion of justice, tax fraud, unlawful association, and irregular funding of the political party in power at the time by businessmen who received benefits for their contributions. One of the people convicted was the wife of the ringleader, the former treasurer of the political party, for being an accessory to some of the crimes committed by her husband and helping him in his criminal activities[7], and another was the former wife of one of the businessmen who made donations to the former. The third and final judgement which should be considered in this context is that of the Supreme Court of 8 June 2018 (the "Noos Case"), which among other individuals punished the former president of an Autonomous Community, a well-known businessman and the brother-in-law of the king of Spain for the crimes of perversion of justice, embezzlement, forgery, public fraud, influence peddling, and tax evasion. Although the wives of the businessman and the king's brother-in-law were not given custodial sentences, they were condemned for benefitting from their husbands' activities (Davies 1999).

All these cases have several things in common: None of the women were convicted of a crime they had committed alone, but rather of being the accomplice of a man on whom they depended or had depended on emotionally, and in many cases, the crime had led them to end their romantic relationship; their participation in the offence occurred once the latter was already established, and thus they were not primarily, but secondarily responsible; and they were basically convicted of money laundering, which can be seen as a "very feminine behaviour", that of "washing" the profits obtained by their partners. The judgements also reveal that in many cases, these women were given more lenient sentences (shorter sentences or lower fines), indicating a certain level of judicial gallantry towards elegant female offenders[8]; their offences were motivated by avarice because they had expensive lifestyles that they desired to maintain—and even enhance—at all costs, and their defence was often based on a "theory of love", according to which wealthy women in love do not look at what they are signing or question where the money they enjoy comes from, but instead show blind, obedient, and trusting love like that between Romeo and Juliet[9].

---

[7]   Two other women were convicted in this case, both administrators of the companies involved in the corruption.

[8]   In many cases, although the court finds that women have committed the same crimes as the men they conspired with, whether for love or work, most of the time they are handed a more lenient sentence. This is the case both when determining the duration of a prison sentence and when calculating the fine to pay in lieu of a prison sentence, since regardless of a couple's economic status, husbands always receive a harsher sentence than wives.

[9]   From a feminist point of view, this defence is based on a model of romantic love in which women do not question their husbands' (economic) decisions because they are uninterested in such affairs due to their gender. Furthermore, according to the traditional distribution of roles in the home, it is the man who is responsible for "these matters". The same is true when a couple works together: Wives basically provide fronts for their husbands, who are not afraid to put their safety at risk. As regards the model of love, one can distinguish between women who claim not to have questioned their husbands' decisions, and those who attempt to show that they were responsible for the couple's economic affairs.

These judicial brushstrokes regarding this new category of criminality lead to the conclusion that there is a very important relationship between pink-collar criminality and gender: The women commit the offence in complicity with a man who holds sway over them in their private life or work. Furthermore, in view of the lifestyle their offences enable them to maintain, they are comfortable with gender stereotypes. Hence, they are not considered transgressive in the same way as women who kill, steal at knifepoint, or defraud vulnerable people: Instead, they are feminine and wish to remain so.

However, for this very same reason, prison will be much harder for these women than for other female convicts, who due to their clothes, behaviour, and way of being will blend into the prison environment and pass unnoticed: Pink-collar offenders must give up the comfort of a wealthy life in which their every whim was met. In short, pink-collar offenders suffer much more in prison than other women because they are unprepared for it, and much more than men because they are more vulnerable to gender stigma, feeling ashamed that they are failing in their family duties as women.

## 4. Prison Treatment of Gender

In Spain, women convicted of drug trafficking, terrorism, or crimes against property receive the same prison treatment, the design of which is based on traditional gender stereotypes whereby women are poor and have economic problems, as is often the case with female inmates. From a gender perspective, this may constitute an equitable measure; however, from a class perspective it does not serve to "treat" all female inmates.

At present, the General Prisons Act and its regulations establish equal conditions for men and women serving their prison terms; furthermore, Royal Decree 1836/2008, of 8 November, abolished the previous separation between male and female prison staff, establishing criteria for integrating the former male and female scales for prison guards. Only a few specialities remain that are related to the sex of the inmate. For example, the basic principle of separation by sex when serving a prison sentence (art. 8.3, 16.a General Prisons Act) establishes that women's prisons must have medical units endowed with the obstetrics equipment necessary to treat pregnant, post-parturient, and nursing inmates and to attend urgent deliveries that cannot be referred to general hospitals (art. 38 of the General Prisons Act and 209 of its regulations). Regarding the disciplinary sanctions to impose, art. 43.3 states that solitary confinement shall not be applied "*to pregnant women, women up to six months after the end of pregnancy, nursing mothers or mothers accompanied by their children*" (see also art. 254 of the regulations). Besides these provisions, there is no other legal differentiation whatsoever.

However, there are two specific sex-related provisions that in the framework of a legal system with an Organic Law, 3/2007, on equality between men and women, should perhaps be reconsidered. The first of these is contained in art. 38.2 of the General Prisons Act and 17 of its regulations, which provide for the possibility that inmate mothers may be accompanied by their children until these reach three years of age (Yagüe Olmos 2002; Naredo Molero 2007). Since this right is only recognised for female—but not male—inmates with children, it seems to perpetuate the notion that it is women who are responsible for child care. It should be noted that art. 99.3 of the regulations and Chapter III of Title VII (art. 68) provide for the possibility of mixed prisons: In these cases, when both men and women are imprisoned and the latter are accompanied by children under three, fathers can be with their children, albeit through the mother (Yagüe Olmos 2002; Naredo Molero 2007).

The second concerns the provision contained in art. 82.2 of the Prison Regulations, which establishes special access to the open regime that is restricted to "*women classified as third-degree* [semi-custodial], *when it has been proven that they will be unable to perform paid work outside, but there is evidence, following the corresponding social services report, that they will effectively carry out domestic tasks in their family home*". This provision, whereby women already classified as third-degree prisoners can be transferred to an open regime, could undoubtedly also be affecting classification as third-degree itself. In both cases, these provisions are based on the traditional distribution of roles, whereby women are responsible for child care and domestic tasks.

It should be noted that in the programme of actions for equality between women and men in prison, one of the recommended measures that the authorities should implement to mainstream gender in prison sentences is as follows: "*in this respect, in light of women's negligible danger to society and in accordance with the principles of the minimum intervention of the law, legal measures are required that mitigate, suspend or replace prison sentences for pregnant women and women with dependent children or adults (these measures should also apply to men with similar family responsibilities)*[10]". Although the goal is laudable, there are other means to benefit these women. For example, regardless of the perpetrator's sex, prison sentences in general now play a less important role in the Penal Code, because alternative punishments are imposed, fewer custodial sentences are handed down for property and public health offences (which still account for the greater part of both the male and female prison population), and there are substitution mechanisms for short prison sentences. Within this framework, if the aim is to incorporate a non-discriminatory gender perspective that does not perpetuate sexist behaviour patterns, one option would be to provide training for women in non-traditional professions and oblige men to assume family responsibilities, showing them that child care, for example, is not the sole responsibility of mothers, but also of fathers. Such a solution would be consistent with the spirit and the letter of the Organic Law on Equality.

However, paradoxically, the training provided for women inmates tends to perpetuate sexist patterns of behaviour by focusing on occupational activities associated with their traditional tasks. Thus, in a typical institution, the workshops available in men's units (e.g., construction, gardening, or electronics) are usually different to those available in women's units (e.g., fashion, film, reading, sewing, macramé, or hairdressing)[11]. The good intentions of correctional institutions in this respect cannot be denied: However, the effectiveness of measures such as these that reinforce the roles assigned by patriarchy to men and women is questionable[12], since they preclude the possibility of learning other kinds of skills. In this respect, Viedma Rojas and Frutos Balibrea have noted that prison work also perpetuates customary sexist limitations. This is related to the type of work women inmates carry out in prison, which is less skilled and worse paid than that of men, leading them to conclude that "*these results suggest that women work more than men, but in the worst tasks. Second, their work mainly involves care (kitchen, laundry) and cleaning*" (Viedma Rojas and Ballibrea 2012).

This finding should be contrasted with the programme of action for equality between men and women in prison, released by correctional institutions, which states that women, "in order to access highly precarious professional activities in the Spanish labour market, such as clothing and crafts, rarely participate in work and activities traditionally considered male", perhaps because the real question is whether women are given the opportunity or encouraged to participate in the latter[13] (Medlicott 2007).

This critique of the sexual division of labour in prison is even more germane when considering the mixed prisons referred to in art. 99.2 of the Prison Regulations or the groups in therapeutic communities referred to in its art. 115.

---

[10] Secretaría General de Instituciones Penitenciarias, "Programa de acciones para la igualdad entre mujeres y hombres en el ámbito penitenciario", cit., p. 37.

[11] Note that as a result of the economic crisis, many workshops and occupational therapy programmes have been discontinued.

[12] http://www.rtve.es/noticias/20110215/festival-ellas-crean-lleva-cine-musica-moda-hasta-carceles-mujeres/407221.shtml. The Association for Human Rights in Extremadura, for example, found that women "*in prison in Extremadura are marginalised with respect to occupational, sporting, recreational, cultural and training activities because the population is overwhelmingly male and there are no specific spaces for women (men leave their units to participate in activities; women do not). In addition, their economic situation is worse because they tend to have children in their care*" (http://centroderechoshumanos.com/acerca-de/adhex).

[13] *Secretaría General de Instituciones Penitenciarias, Programa de acciones para la igualdad entre mujeres y hombres en el ámbito penitenciario*, op. cit., p. 26.

## 5. Conclusions

The specific provisions for women inmates contained in Spanish prison legislation perpetuate existing patterns of behaviour in society as a whole, attributing sole responsibility for care of the home and children to women. Given the impact of these same gender stereotypes on female criminality, one might conclude that the formal control of criminal law relies on the informal control of gender roles to organise prison life. Combined with the United Nations' finding that a high number of women inmates have been subjected to violence by their husbands or partners, one could conclude that it is contradictory to provide a prison "treatment" that is based on the same gender roles that have governed women's entire lives.

This in no way implies that all women offenders are victims, which would be a victimhood interpretation of gender-related female criminality[14], but it does underscore the fact that when victimisation and criminality coincide, they are both cause and consequence through a relationship of causality. Hence, not all women who have committed an offence have also been victims of violence at the hands of a man, but some have. In the end, it is possible to establish a relationship of causality from the point of view of gender stereotypes between primary victimisation-criminality and secondary victimisation in prison. Thus, policies aimed at preventing victimisation also help prevent female criminality (or at least one of the forms of female criminality).

It seems more probable that the way to prevent female criminality is to prevent victimisation and eliminate gender stereotypes (Acale Sánchez 2017, p. 1). The process of social reintegration of these women will entail the launch of treatment programmes that not only prepare them for release, but do so from the perspective of an egalitarian and hence fairer society. Consequently, prisons cannot ignore the impact of gender on female victimisation and criminality, especially in cases where the offence was preceded by an act of victimisation, because otherwise they will be ignoring something that is of major importance in these women's lives. It is of little use to denounce gender-based violence as the most brutal form of discrimination against women while at the same time offering sexist, unequal treatment programmes; the State is obliged to remove any obstacles that prevent women inmates from serving their sentences under equal conditions to men, highlighting the measures that exacerbate the harshness of their lives in prison and the lives they left behind, pursuant to the provisions of art. 9.3 of the Constitution.

Society as a whole plays an essential role in the criminalisation and social reintegration of these women, who are often condemned more for the sin of having offended than for the fact of having committed a crime, and who, rather than receiving a punishment, are given a "sentence" loaded with social prejudices that render it harsher than when imposed on a man. In short, if we consider the verb "to punish", we could say that society cannot limit itself to contemplating how these women are "being punished".

As I said at the beginning, the present study concludes a research project focusing on female criminality in Spain, a country that maintains cultural and political relations with the European Union, the Council of Europe, and the United Nations, and has the same vision of the problem of female criminality as many other countries. In the final analysis, female criminality is a structural phenomenon of the patriarchal system, which cannot be controlled by domestic penal legislation, but by addressing the patriarchal laws that govern the behaviour of men and women throughout the world, as evidenced by the case in Spain.

**Funding:** This research received no external funding.

**Conflicts of Interest:** The author declares no conflict of interest.

---

[14] In particular, as regards immigrant women, Ruth Mestre i Mestre (2005) has observed that there is a *"tendency to consider them* [women] *victims of immigration law, the patriarchy, mafia networks, institutions and a long list of variations on the same theme"*, highlighting the strategies aimed at *"affirming their capacity for action and decision-making"*.

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
