# Peer review of "Penal and Custodial Control of Female Criminality in Spain from a Gender Perspective"

_socsci, doi:10.3390/socsci8020052_

Round 1

Reviewer 1 Report

Some references in the introduction could help to situate the international discussion on the topic. The second section begins with only Spanish references and information, but it is not clear from the introduction why the Spanish case is interesting, nor in what moment the author begins to explore the Spanish context and not the international one. It would be interesting to contrast the results with the international situation (not necessarily the Anglo-American one). To compare with other European countries could be informative. Is the Spanish situation normal in an international context? 

The title should be changed. It does not reflect the fact that the article is about Spain.

Author Response

I want to thank the reviewer for the observations. I don’t make any more references about other countries because I am only studying the Spanish system and I want to publish my paper in a specific issue of Social Sciencies about “gender, crime and criminal justice”. I am waiting for other people to write about other countries in this issue. In any case, in the introduction and in the conclusion I explain why I only study the Spanish system. I have also introduced some references to Anglo-Saxon literature. Now, I think it is possible to say that the recomendations have been included.

Reviewer 2 Report

The paper is oriented towards the analysis of a subject that has been little studied in Spain, as well as in the context of other southern European countries so far, and therefore deserves to be published.

However, although female criminality and the stereotypes on which it is based have been scarcely analysed in the country in which this research has been carried out, it is an issue profusely analysed in Anglo-Saxon literature. Consequently, It would be desirable for the article to put the object of the research into a more international context, above all in the introduction, although also relating the findings of the Spanish study with the results of previous studies on this issue. In order to do that, it would be necessary to carry out a literature review on the most remarkable literature issued on this topic in English introducing it in the paper.

On the other side, although I am not an English native speaker, it can easily be seen that the quality of the article would also improve with an in-depth English edition.

Author Response

I want to thank you for these observations: they are the same as the observations from reviewer 1. As I have just said, I don’t want to study other countries because I want to publish this paper in a special issue of Social Sciencies about Gender, Crime and Criminal Justice.  I am waiting for other people to write about gender, crime and criminal justice in America or in other countries in this issue.

I am sorry but I have contrated a professional legal translator to translate. I hope everything is correct.

Round 2

Reviewer 1 Report

Thank you for paying such an accurate attention to my observations. Proof-reading would improve the quality of the language.